# Correlation of Statistical Distributions of the Dimension of Yeast Cells Attached to the Substrate and Its Surface Electrical Potential

**DOI:** 10.3390/ma15010006

**Published:** 2021-12-21

**Authors:** Elina Bondareva, Yuri Dekhtyar, Vladislavs Gorosko, Hermanis Sorokins, Alexander Rapoport

**Affiliations:** 1Riga Technical University, Kaļķu Street 1, LV-1568 Riga, Latvia; elina.bondareva@gmail.com (E.B.); jurijs.dehtjars@rtu.lv (Y.D.); vladislavs.gorosko@gmail.com (V.G.); 2Laboratory of Cell Biology, Institute of Microbiology and Biotechnology, University of Latvia, Jelgavas Street 1, LV-1004 Riga, Latvia; aleksandrs.rapoports@lu.lv

**Keywords:** *S. cerevisiae*, cell adhesion, cell sizes, surface electrical potential

## Abstract

The ability of cells to adhere to substrates is an important factor for the effectiveness of biotechnologies and bioimplants. This research demonstrates that the statistical distribution of the sizes of the cells (*Saccharomyces cerevisiae*) attached to the substrate surface correlates with the statistical distribution of electrical potential on the substrate’s surface. Hypothetically, this behavior should be taken into consideration during the processing of surfaces when cell adhesion based on cell size is required.

## 1. Introduction

Yeast cells *Saccharomyces cerevisiae* are widely used in various biotechnological applications such as bakery, brewing, medicine, ethanol production, etc. [1,2]. The application of *S. cerevisiae* (*Sc*) for the production of ethanol has been of particular interest to researchers since the mid-2000s. This becomes evident after evaluating the annual number of publications on this topic, with the number of articles having the keyword pairs “*S. cerevisiae*” and “ethanol production” eclipsing those related to bakery, brewing or medicine (Figure 1).

Ethanol yield can be increased by immobilizing *Sc* cells on the surface of a substrate [3,4]. An increase in microscale roughness and porosity also increases the area available for cells to attach. Therefore, immobilizing cells on a substrate with adequate microscale features may lead to increased ethanol yield when compared to free-floating cultures. To make the substrate suitable for cell attachment and proliferation, different surface functionalization approaches can be employed: modification of substrate surface morphology, plasma treatment, the application of various coatings [5,6,7], as well as electrical charge deposition [8,9,10,11]. However, first of all, the cell must be immobilized on the surface. For this purpose, surface functionalization approaches that employ purely physical means should be considered first and foremost since the primary interaction between the cell and substrate surfaces is governed by the same physics of adhesion as are the interactions between any inanimate particle or surface [12].

As prescribed by [12], the particle—a cell, in this case—may adhere if the particle-surface interaction energy has a minimum situated at the surface. This minimum is a result of the superposition of the attractive and repulsive energies that influence the cell which are expressed as the van der Waals (vdWP) and Coulomb (CP) potentials, correspondingly. It is generally known that vdWP decreases very fast with a distance x (~x^−6^) compared with the reduction of CP (~x^−1^) [13]. Thus, to control the attachment of cells, particularly those that are located at long distances from the surface, the use of phenomena that compound CP may be more effective than those that compound vdWP. The presence of an electric charge—which is the phenomenon directly responsible for CP—on a surface of a dielectric substrate can, therefore, affect the attachment of cells [9,10].

During the biotechnological step of *Sc* cell immobilization from a liquid medium onto a solid substrate surface, the cells are homogeneously distributed in a solution, which flows over the substrate surface in a layer thicker than the substrate [3,4]. In this case, CP penetrates further through the medium than vdWP, hence the effects of CP are extended to a higher number of cells able to attach to the substrate surface.

Because of the electric charge present on a dielectric substrate’s surface, the gradient electric field *dE*/*dx* (*E*–intensity of the electric field) influences the motion of the cells in the solution [14]. Therefore, the electrostatic force (*F*) that acts on the cell dipole is [14]:$$F=p\frac{dE}{dx}$$
with *p* being the dipole moment of a cell, expressed as:p=ql
*q—*electric charge of the dipole point charges,*l—*distance between the point charges; *l* is directly linked to the size of the cell [15].

In fact, since *Sc* cells are ellipsoid-like in shaped [16], the value of their dipole moment is directly influenced by their dimensions [15]. Therefore, if the goal is to attach the highest possible number of cells to the substrate surface and since the *Sc* cell dimensions vary among their population [17], the electric field at the substrate surface must be statistically distributed over it. However, this assumption has not yet been proven.

The article aims to explore the correlation between the statistical distribution of the immobilized *Sc* cell dimensions and the distribution of the substrate surface electrical potential.

## 2. Materials and Methods

### 2.1. Materials

Glass slides (Thermo Scientific, Menzel, Germany) [18] were used as substrates to be electrically charged and undergo cell deposition onto their surface. The main component of glass, an amorphous, dielectric material, is SiO_2_. These three factors promote the formation of electron traps (dangling bonds, interstitial defects [19], etc.) both in the bulk and on the surface of the substrate [20]. The filling of such traps with electrical charge carriers can lead to surface charging [21,22].

Glass strips were cut by hand with a diamond scriber into 10 × 10 mm^2^ individual substrates further used during cell deposition. The surface morphology of the substrates was tested using an atomic force microscope (Solver–P-47 Pro, NT-MDT, Chernogolovko, Russia) performed in tapping mode. NSG10/Pt scanning probes (NT-MDT Spectrum Instruments, Moscow, Russia) were employed. The roughness was characterized as a value of Ra measured at the base of 10 nm. A representative image of an AFM surface roughness scan performed on glass substrates is given in Figure 2.

The roughness of each substrate was measured randomly on the selected 5–7 areas. Initially, the scans were to be 5 × 5 µm^2^ sized squares. However, this scanning was time consuming. Therefore, to save time, the scanned area was decreased to a 1 × 1 µm^2^ square. To verify that 1 × 1 µm^2^ is the equivalent of 5 × 5 µm^2^ area the statistical distribution of Ra for both squares were compared using the Kolmogorov-Smirnov (KS) test [23]. As a result, the 1 × 1 µm^2^ area represents the 5 × 5 µm^2^ with uncertainty 5% (*p* = 12).

The Ra was equal to 10 ± 1 nm, that is significantly less than the size of the yeast cell 1–50 μm [24]. Therefore, the substrate surfaces were designated as smooth.

### 2.2. Electrical Charge Provision to the Substrates

Electrical charge deposition on the substrate surfaces was performed using ultraviolet radiation (UV) [25,26] with a broad-spectrum xenon-mercury UV lamp (Lightningcure LC5, Hamamatsu, Japan). The intensity of the light source was around 3500 W/cm^2^ at the source, and the radiation stability was within ±5% according to the lamp’s certificate provided by the manufacturer. The distance between the light source and the substrates surface was maintained at 30 cm to keep the substrate from heating up during exposure, and the exposure time was 30 min [25,26].

To achieve the specific distribution of the electrical charge over the substrates surface, the UV beam was delivered at defined angles (α), those being 30°, 45°, and 60° between the beam and the substrate surface.

The statistical distribution of the electrical charge was identified by measuring the surface electrical potential of the substrates, which was performed via Kelvin probe force microscopy (KPFM) [27]. The previously mentioned atomic force microscope and scanning probe were used. The surfaces were scanned with the lateral resolution of 30 nm. KPFM measurements on 24 randomly selected areas measuring 5 × 5 μm^2^ each were performed for all samples. A representative of a KPFM scan performed on a glass surface after exposure to UV light is given in Figure 3. The processing of acquired data was performed using Origin Pro (version 2019b, OriginLab Corporation, Northampton, MA, USA).

### 2.3. Cells and Their Immobilization

Dried yeast *S. cerevisiae* were used in the inactive form. The cells were acquired from the Rigas Raugs Company (Jästbolaget AB, Sollentuna, Sweden). Before deposition, the cells were mixed with distilled water at an amount of 0.5 g yeast powder per 100 mL of water. The solution was then diluted with distilled water to reach a cell concentration that corresponds to an optical density (OD) of 0.15 ± 0.05 units dat a probing wavelength of 600 nm that is generally used to identify cell concentration [28]. The value of the optical density was selected following [26]. Optical density measurements were performed using the Spectronic Helios Gamma UV-Vis (Thermo Fisher, Waltham, MA, USA) spectrometer.

Before cell deposition, the substrates were consequently rinsed in acetone, isopropanol, ethanol, and distilled water to remove surface adsorbates [26].

After that the substrates were finally cleaned using an ultrasonic bath—Bandelin Sonorex electronic RK31. For this, the substrates were put into a plastic jar which was placed into the ultrasonic bath. The substrates were processed in an ethanol–distilled water solution (1:1) for 15 min at an ultrasound frequency of 35 kHz. The procedure was repeated twice to ensure that all samples were cleaned thoroughly. This was identified using the Carl Zeiss Jena NU-2 (Carl Zeiss Meditec AG, Jena, Germany) optical microscope in reflective mode with the magnification set to 125×. Finally, the substrates were dried in room conditions. The described cleaning protocol was used on all samples both before AFM measurements and before *Sc* cell deposition.

An orbital shaker (OS-20, Biosan, Riga, Latvia) was used to immobilize the cells on the substrates. The substrates were placed in a Petri dish and were completely coated with suspension. The Petri dish with the suspended cells was placed on a shaker and was mixed for 60 min at 30 rpm. After shaking, the substrates were kept stationary for 30 min to let the cells settle down. Then the suspension was removed, and the samples were left to dry in air for 20 min at 30 °C.

During the whole period of cell deposition, the level of suspension above the substrate surface was kept at about 0.5–1 mm, which was checked with a caliper. This allowed the cells to deposit in a single layer, preventing possible overlays of charged areas with several cell layers, which would shield the charge. In this case, identification of the influence of charge distribution on the substrate surface is impossible.

The distribution of cells over the surface of the substrates was imaged using a Carl Zeiss Jena NU-2 (Jena, Germany) optical microscope in reflective mode with the magnification set to 125x. The images were recorded with a digital camera attached to the microscope. Fiji (ImageJ 1.53e, Wayne Rasband and contributors National Institute of Heath, USA) software was used to measure the sizes of the cells [29] by utilizing the *Straight* function to draw a straight line between the two furthest points of a cell and then the *Measure* function to determine the length of the drawn line in pixels. The acquired pixel lengths were recalculated into micrometers using a measurement of the bar acquired using the same approach. OriginPro 2019 was used for the statistical analysis of the acquired data. Images taken from the surfaces of 24 substrates were used. A sample image of immobilized *Sc* cells used during analysis is given in Figure 4.

## 3. Results and Discussion

Table 1 shows the average values of surface electrical potential for substrates exposed to UV light at different angles. It indicates that changes in the exposure angle influence the charge deposition with electrical potential values being at their lowest at 30 degrees and at their highest at 45 degrees.

Figure 5 demonstrates the relationship between the average size of attached cells and the average value of substrate surface potential. This suggests that the average size of attached cells gradually increases with an increasing surface electrical potential.

The histograms given in Table 2 depict the distributions of the substrate electrical potential values and the attached cell sizes. Frequency and cell size frequency axes depict the probability of finding an electrical potential or a cell size value within a dataset. When comparing native, nonexposed samples to their exposed counterparts, nonexposed substrates have only one pronounced maximum, both for distributions of electrical potential and attached cell size values. Exposed substrates, however, in both cases exhibit multiple maxima, regardless of exposure angle.

Figure 6 shows that the resulting histogram curves can be approximated with two gaussian curves. The relative height of these curves changes gradually for surface electrical potential values, with one of the two peaks decreasing in prominence with an increasing exposure angle and vice versa occurring for the other peak. The attached cell size distribution, however, demonstrates a different behavior with the second peak decreasing in height and then returning to a similar value over the 30–60 degree range.

Overall, the results show that surfaces with lower values of electrical potential are more conducive to the attachment of smaller *Sc* cells while higher potentials attract larger cells. Furthermore, different modes of surface charge distribution affect the size distribution of the attached *Sc* cells.

## 4. Conclusions

The average size of yeast cells (*S. cerevisiae*) attached to the substrate surface increases with its electrical potential average value.The statistical distribution of the sizes of the yeast cells (*S. cerevisiae*) attached to the substrate surface is in accordance with its surface electrical potential statistical distribution.The results may be used to:
Sort cells by cell size;Process the surface of biosubstrates to achieve the high distribution of differently sized immobilized cells.


## Figures and Tables

**Figure 1 materials-15-00006-f001:**
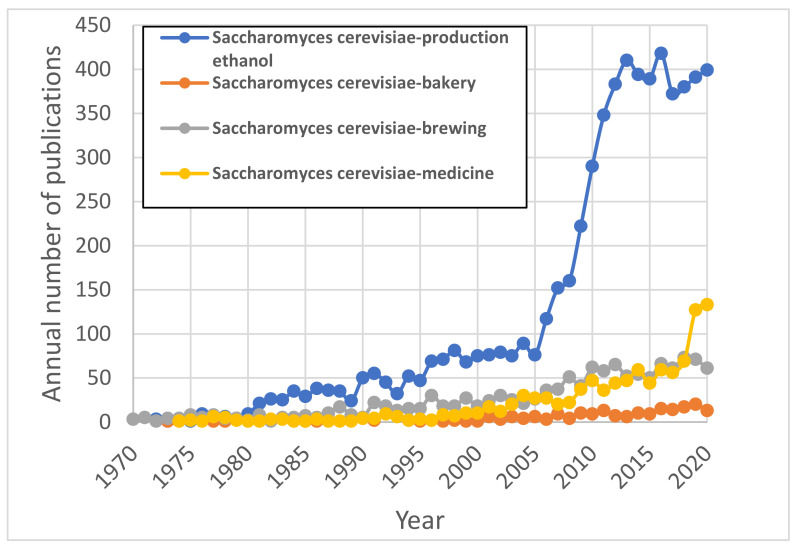
Annual number of publications of *Sc* applications in different areas (the legend indicates the keywords that were used when performing the SCOPUS engine search).

**Figure 2 materials-15-00006-f002:**
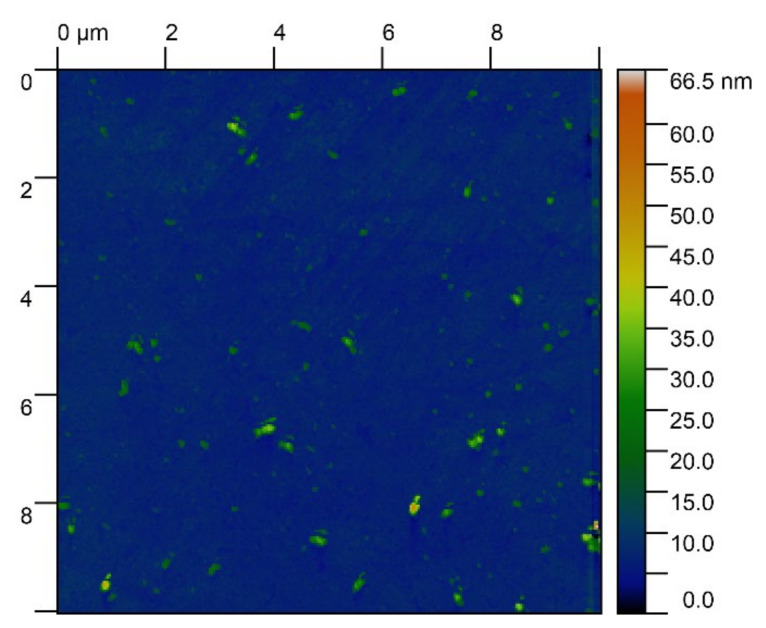
Representative image of surface roughness for soda-lime glass samples.

**Figure 3 materials-15-00006-f003:**
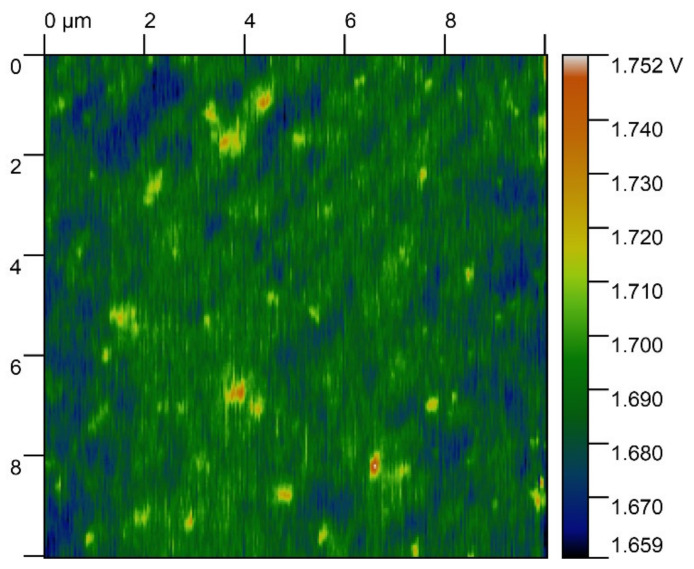
Representative image of surface electrical potential for sod-lime glass samples after irradiation at a 60-degree angle.

**Figure 4 materials-15-00006-f004:**
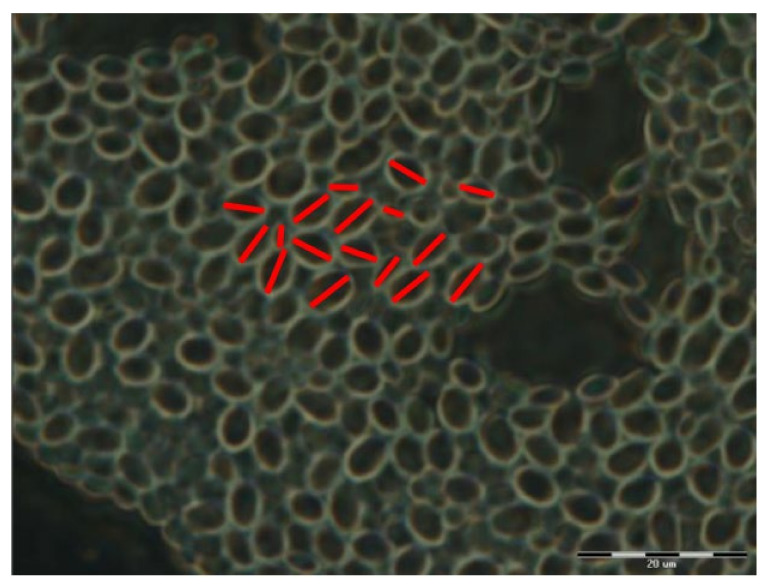
*Sc* cells deposited on the surface of a substrate. Similar images were used for processing. Red lines depict the approach used for measuring cell size.

**Figure 5 materials-15-00006-f005:**
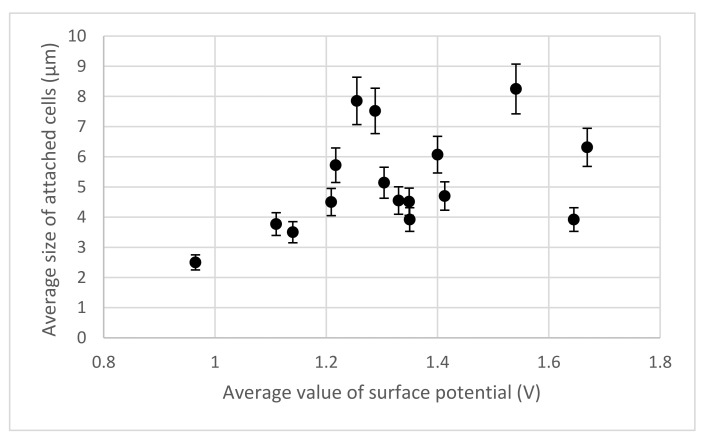
The correlation of the average value of surface potential and the average size of the attached cells (the vertical whiskers correspond to the standard deviation of the average size of the attached cells).

**Figure 6 materials-15-00006-f006:**
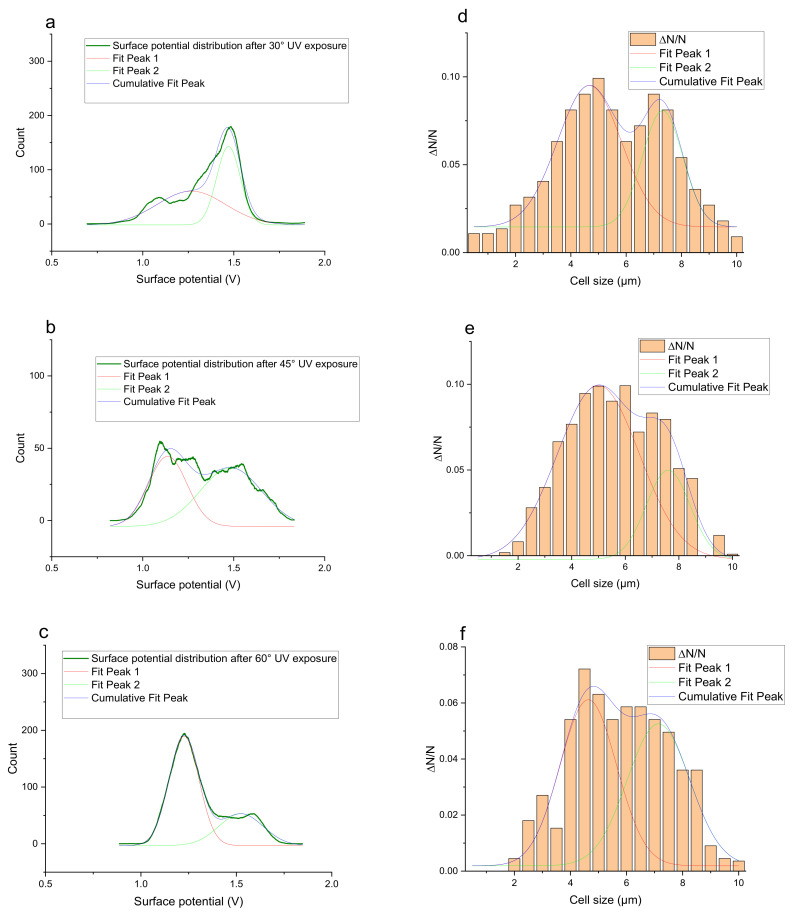
(**a**) Surface potential charge distribution of sample irrigated from 30° angle. (**b**) Surface potential charge distribution of sample irrigated from 30° angle. (**c**) Surface potential charge distribution of sample irrigated from 30° angle. (**d**) Attached yeast cell size distribution on samples irrigated from 30° angle. (**e**) Attached yeast cell size distribution on samples irrigated from 45° angle. (**f**) Attached yeast cell size distribution on samples irrigated from 60° angle.

**Table 1 materials-15-00006-t001:** The values of average surface potential (P), dispersion of surface potential values (Dp), and their ratio (P/Dp) in relation to the exposure angle α.

Angle α	P, V	Dp, V	P/Dp
30	1.39 ± 0.05	0.063 ± 0.005	22.06 ± 0.1
45	1.72 ± 0.08	0.135 ± 0.017	12.74 ± 0.47
60	1.65 ± 0.07	0.084 ± 0.009	19.64 ± 0.77

**Table 2 materials-15-00006-t002:** Histograms of the surface electrical potential and immobilized cell size distributions.

Substrate Type		Distribution of Electrical Potentials, V	Distribution of Attached Cell Sizes, μm
Native substrate		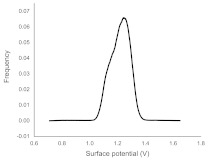	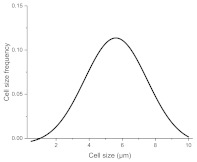
Irradiated substrate	α = 30°	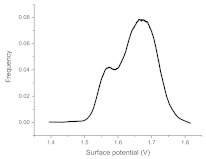	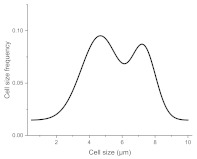
	α = 45°	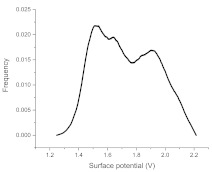	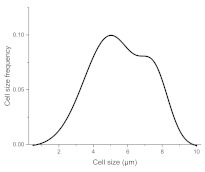
	α = 60°	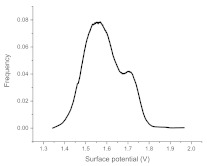	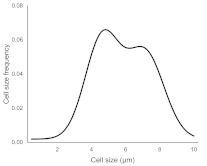

## Data Availability

The data presented in this study are available on request from the corresponding author.

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
