# Peer review of "Correlation of Statistical Distributions of the Dimension of Yeast Cells Attached to the Substrate and Its Surface Electrical Potential"

_materials, 2021, doi:10.3390/ma15010006_

Round 1

Reviewer 1 Report

The manuscript "Correlation of statistical distributions of the dimension of yeast cells attached to the substrate and its surface electrical potential" by Bondareva et al. describes the correlation of cells adherence (S. cerevisiae cells) and the electric surface potential of the substrate.

In general, the works carves that cells size and adhesion tendency scales, meaning that larger cells adhere at higher potentials. Unfortunately, this is the only outcome of the manuscript.

Other factors such as cell potential or environmental conditions are left undiscussed and unattended.

The manuscript is extremely short and is in my opinion not a full article, but rather a technical report. For example, other cells types should be included and the type of substrate and the attachment conditions should be investigated in much more detail to make it a full article.

In the current status of the manuscript, I cannot recommend its publication in "Materials".

Minor comment: strain names are written in italics.

Reviewer 2 Report

The manuscript entitled ‘Correlation of statistical distributions of the dimension of yeast cells attached to the substrate and its surface electrical potential’ presents some aspects of the influence of electrical potential on the distribution and size of yeast cells on the substrate. The idea is interesting and it may provide valuable information on the effect of the electric current on fungus microorganisms. However, this is not enough to claim that the process will be identical and have similar results in the case of mammalian cell populations used in in vitro tissue engineering studies. Other observations are:

  1. There are no sufficient references to sustain the scientific background and the findings. At least 25 bibliographic references should be added. Also, the mentioned references are not recent enough and the format is not appropriate (i.e., reference ‘7’).
  2. The introduction is insufficient and refer to bioimplants and biotechnologies which are obviously associated to human/ mouse/ rat/ rabbit cells, while the manuscript presents and investigation on fungus cells.
  3. It is affirmed that ‘Typically, the attachment is controlled by van der Waals and detachment by Coulomb forces according to a reference [K. Gayen, K. v Venkatesh, Quantification of cell size distribution as applied to the growth of Corynebacterium glutamicum, 165, Microbiological Research. 163 (2008) 586–593. https://doi.org/https://doi.org/10.1016/j.micres.2006.07.017]. More recent publications indicate the complexity of cell adhesion process and how this is integrin regulated [Robert Philippe, et all (2021) Functional Mapping of Adhesiveness on Live Cells Reveals How Guidance Phenotypes Can Emerge from Complex Spatiotemporal Integrin Regulation, J. Frontiers in Bioengineering and Biotechnology, Vol. 9, pp. 226; Yogambha Ramaswamy et al (2021) Nature-inspired topographies on hydroxyapatite surfaces regulate stem cells behaviour, Bioactive Materials, Volume 6, Issue 4, Pages 1107-1117]. There is no doubt that the electrical charge influence cells adhesion to a substrate, which has been already proved by many [e.g., Diana V. Portan et al (2019) Combined Optimized Effect of a Highly Self-Organized Nanosubstrate and an Electric Field on Osteoblast Bone Cells Activity, BioMed Research International, Volume 2019, Article ID 7574635, 8 pages]. However, this is not the key parameter since surface nature, properties, texture of the substrates and how these regulate integrins in cells is crucial.
  4. Why was 'glass' the only used substrate? There is no control material such as TCP (tissue culture plastic) to validate the results in the absence of any stimuli (electricity). Finally, glass is a very poor conductor of electricity.
  5. The manuscript is rather a technical report on how to electrically charge some glass substrates. Interpretation of results is poor. Biomarker analyses were not performed at all, while all biological aspects are neglected.

Reviewer 3 Report

The manuscript “Correlation of statistical distributions of the dimension of yeast cells attached to the substrate and its surface electrical potential” presents a relation between the distribution of electrical potential of UV treated glass substrates with the distribution of dimension of yeast cells adhered to the substrates. As the authors mention, the results presented may be of interest for the development of  new implants and bio surfaces. I find very interesting the use of Kelvin prove force microscopy (KPFM) for the spatial characterization of surface electrical potential. However, I think the manuscript needs great improvement before being considered for publication.

-I think both the abstract and the introduction of the manuscript should be improved. The abstract can be improved adding some information regarding the experimental part, for example, by mention of the use of KPFM. The introduction can be also improved explaining why controlling cell adhesion characteristics is important in the field of implants and bio surfaces. Furthermore, why the authors choose to use Saccharomyces Cerevisiae as a cell model?

-Why is it important to show the chemical composition of the substrates in a table?

-Is it possible to include some of the AFM images that were used to obtain the surface electric potential distributions? Is it possible to include some images of the yeast adhered to the different substrates that were used for cell size quantification?

-Is it possible to discuss the effect of the surface electric potential on yeast size in terms of their cell wall composition and charge?

-How the authors define ‘cell size’? Diameter?

-Some cells are larger than the 5x5 square micron areas measured with the KPFM. Is that a problem for drawing conclusions?

-What is the potential range plotted in figure 2?

-Some superscripts are missing (for example lines 57 and 69) and some weird symbols appear in for example lines 63 and 69.

-In table 3, the histograms presenting  cell size frequency have a label that says ‘Histogram of the potential’.

-Was the cleaning protocol from section 2.3 also applied before the KPFM measurements?

Reviewer 4 Report

The authors present a study on saccharomyces cerevisiae cell size distribution of surface with different potential.

Although the experimental section is well presented, the contextualisation of the study is very poor. The author open the introduction focusing on the importance of cell adhesion to bioimplants. In this case, I would have expected the use of mammalian cells for investigating their adhesion. Moreover, this affirmation is not always true. Some implants indeed are functionalised to avoid any interaction with cells or proteins. In addition, it is known that increasing surface potential increase cell adhesion.

I think that the work can be of interest in the light of using the strategy to select specific cell sizes. However, improvement with respect to the state of art should be better highlighted. The introduction, discussion and the reference number are indeed too much short. 

Results: error bars are missing in Fig 2 and 3. Statistical analysis should be presented. 

Round 2

Reviewer 1 Report

The manuscript has been substantially improved. However, i still think that the publication type "article" is not appropriate for this work. It should be rather called a technical report or short communication.

Except from this aspect, i can recommend the publication of the manuscript in "Materials".

Author Response

Dear reviewer,

Thank you for your comments during the second round of the review process. The authors would like to extend their deepest gratitude to the reviewers that considered the updated manuscript worthy of being published and to all who have spent their valuable time revising the work. The authors have corrected the few remaining issues that were pointed out.

Sincerely,

Valdislav Goroshko 

Reviewer 2 Report

Dear Author,

Figure 5. The correlation of and the average size of the attached cell. - PLEASE CORRECT LEGEND

No further comments.

Author Response

Dear reviewer,

Thank you for your comments during the second round of the review process. The authors would like to extend their deepest gratitude to the reviewers that considered the updated manuscript worthy of being published and to all who have spent their valuable time revising the work. The authors have corrected the few remaining issues that were pointed out.

Sincerely,

Valdislav Goroshko 

Comment by the reviewer

Comment by the authors and revision of the manuscript

Figure 5. The correlation of the average size of the attached cell. – Please correct legend.

Thank you for the comment. We’ve corrected the description of Figure 5 which now fully describes both the figure and the included error bars.

Reviewer 3 Report

The manuscript has been largely improved after authors revision. The introduction now provides background and support to the described experimental design and results. I appreciate the addition of the AFM and Sc images. I think the manuscript is suitable for publication in Materials.

Author Response

(The authors gave the same response as above.)

Reviewer 4 Report

The authors addressed all my previous comments, improving the manuscript.

Author Response

(The authors gave the same response as above.)
